# Glucose Metabolism Disorders and Parkinson’s Disease: Coincidence or Indicator of Dysautonomia?

**DOI:** 10.3390/healthcare12232462

**Published:** 2024-12-06

**Authors:** Tomasz Chmiela, Przemysława Jarosz-Chobot, Agnieszka Gorzkowska

**Affiliations:** 1Department of Neurology, Faculty of Medical Sciences in Katowice, Medical University of Silesia, 40-752 Katowice, Poland; 2Department of Children’s Diabetology and Lifestyle Medicine, Faculty of Medical Sciences in Katowice, Medical University of Silesia, 40-752 Katowice, Poland; pjarosz-chobot@sum.edu.pl; 3Department of Neurology, School of Health Sciences, Medical University of Silesia, 40-752 Katowice, Poland; agorzkowska@sum.edu.pl

**Keywords:** continuous glucose monitoring, autonomic disorders, glucose metabolism disorders, Parkinson’s disease

## Abstract

**Background:** Parkinson’s disease (PD) and type 2 diabetes mellitus (T2DM) are both age-related diseases. Evidence from recent studies suggests a link between them. The existence of an interaction between autonomic nervous system dysfunction and the dysregulation of glucose metabolism is one of the proposed mechanisms to explain the complicated relationship between these diseases. The aims of this study are to assess the incidence of glycemic dysregulation in people with PD and to identify clinical factors that may predispose patients with PD to the occurrence of metabolic disturbances. **Methods:** In total, 35 individuals diagnosed with PD and 20 healthy control subjects matched in terms of age and gender participated in a study consisting of clinical and biometric assessments along with 14 days of continuous glucose monitoring (CGM) using the Freestyle Libre system. In the group of patients with PD, a comparative analysis was performed between patients with and without autonomic dysfunction. The severity of autonomic dysfunction was assessed using the SCOPA-AUT. **Results**: Participants diagnosed with PD demonstrated a trend toward lower morning glucose levels compared to the control group. PD patients with autonomic symptoms had greater glucose variability and a deeper trend toward lower glucose levels in the mornings. The presence of autonomic dysfunction, especially orthostatic hypotension and micturition disturbance, and the severity of autonomic symptoms were associated with greater glycemic variability. **Conclusions**: The occurrence of autonomic disorders in the course of Parkinson’s disease predisposes patients to more profound glycemic dysregulation.

## 1. Introduction

Parkinson’s disease (PD) and type 2 diabetes mellitus (T2DM) are both age-related chronic diseases, and recent evidence indicates that they share some dysregulation pathways. Diabetes mellitus (DM) is a group of metabolic disorders whose main feature is chronic hyperglycemia. T2DM is the most common form of DM, accounting for about 90% of cases, and it is characterized mainly by insulin resistance [1]. The classical progression of T2DM includes the development of glucose intolerance, then elevated fasting glucose, and finally elevated hemoglobin A1C; such disorders that do not reach the threshold for T2DM are called “prediabetes” [2]. A relationship between T2DM and neurodegeneration has been observed in population studies with people with T2DM shown to have a 66% increased risk of developing Alzheimer’s disease [3] and a 36% increased risk of developing PD [4]. More than 60% of people with PD are glucose intolerant, and insulin resistance was found in 62% of Parkinson’s patients with dementia, of whom 30% were glucose intolerant [5]. Epidemiological data on the co-occurrence of PD and T2DM may indicate a relationship between these two diseases but are unclear [6,7,8]. Long-term diabetes is thought to be associated with an increased risk of degenerative diseases, including PD [9], and diabetes has also been found to influence the severity of both motor and non-motor symptoms of PD [8]. The relationship between neurodegeneration in PD and dysglycemia is likely to be complex and bidirectional [7,10,11,12,13,14,15]. Emerging evidence suggests a common pathological pathway linking PD and T2DM, involving factors such as altered insulin signaling, insulin resistance, oxidative stress, mitochondrial dysfunction, neuroinflammation, and protein accumulation [13,14,15,16]. Interestingly, in type 1 DM, a state of insulin deficiency, insulin sensitivity, and a lack of insulin resistance is not associated with an increased risk of PD [17]. Environmental factors and genetic susceptibility are both likely to contribute to the common pathophysiological mechanism of PD and T2DM [18,19]. Exposure to heavy metals and pesticides is a known risk factor for PD [20]. Similarly, heavy metal exposure may cause islet dysfunction and disease progression in T2DM [21]. Recent studies have shown that PD and T2DM share some genetic susceptibilities that put individuals at risk for both diseases. For example, single-nucleotide polymorphisms in Akt, which encodes AKT serine/threonine kinases, increase the risk of both PD and T2DM [22], or the PD-related Park7 gene encodes the protein DJ-1, which is associated with islet loss in T2DM patients [23].

Based on data from previous studies, it has been hypothesized that mitochondrial dysfunction, endoplasmic reticulum stress, inflammatory processes, and other metabolic changes may lead to insulin resistance and further diabetes and neurodegeneration [6,15,19,24]. It is also known that insulin is an essential regulatory protein for the central nervous system, controlling many important brain functions such as neuronal survival, autophagy of toxic proteins, synaptic plasticity, neurogenesis, oxidative stress, and inflammation [19]. It has also been noted that pancreatic islets are innervated by dopaminergic neurons [25,26], and insulin secretion is regulated by the autonomic nervous system [26,27]. In turn, autonomic dysfunction is a clinical feature of synucleinopathies such as PD [28,29]. Furthermore, in a previous study, we showed that glycemic dysregulation is more common in synucleinopathies such as PD or multiple system atrophy (diseases with frequent autonomic disorders) than in progressive supranuclear palsy, and classic risk factors for metabolic syndrome are not associated with this difference [30]. The potential pathophysiological links between PD and DM are briefly summarized in Table 1.

The aims of this study are to assess the incidence of glycemic dysregulation in people with PD, to evaluate the daily glucose trends and glycemic variability in this population, to assess the importance of clinical features of dysautonomia in parallel with glycemic dysregulation, and to identify clinical factors that may predispose patients with PD without diagnosed T2DM to the occurrence of metabolic disturbances. To our knowledge, this is the first study to include 14-day continuous glucose monitoring (CGM) in a population of people with PD and provide an initial insight into glucose trends in PD. However, due to the single-center nature of this study and the relatively small cohort, the generalization of our results requires further investigation. Multicenter studies, optimally performed on different PD subtypes and patients in different stages of the disease, using objective measures such as PET with 18F-fluorodeoxyglucose or alpha-synuclein measurements, are needed to explore the significance of glucose trends alteration in PD.

## 2. Materials and Methods

Forty-one individuals with confirmed PD according to the 2015 MDS diagnostic criteria [31] were screened for eligibility for this study. The inclusion criteria were a clinically confirmed diagnosis of PD and age over 18 years. The exclusion criteria were a diagnosis of DM (both type 1 and type 2, known from an interview or diagnosed at screening), the use of medications that may affect glycemic levels, or a change in PD therapy within one month prior to examination (as this may affect glucose levels). Patients who had undergone deep brain stimulation (DBS) surgery were excluded because DBS is known to affect glucose levels and may cause weight gain [32,33]. Patients with conditions that could influence their glucose level, such as any acute illness, chronic kidney disease (stage above G4), or diagnosed thyroid disease or another endocrine disease associated with glycemic disorder, were excluded from this study. The inclusion and exclusion criteria are summarized in Table 2.

Prior to screening, all patients were informed of the study objective, protocol, and potential risks. They were assured that the lack of consent would not affect their treatment and that they could withdraw their consent to participate at any time during the study. Only patients who provided written consent were screened. The content form was reviewed and approved by the Bioethics Committee of the Medical University of Silesia.

### 2.1. Screening Procedures

The screening procedures included an interview on the course of the interviewee’s Parkinson’s disease, their current treatment and comorbidities, and a neurological examination. The participants were assessed by a neurologist experienced in movement disorder and were evaluated in terms of meeting the PD diagnostic criteria [31]. Based on their current treatment, the levodopa equivalent daily dose (LEDD) was calculated. The calculations were based on a systematic review by Tomlinsen et al. [34].

Laboratory screening tests to exclude T2DM were performed, including fasting glucose, hemoglobin A1C (HbA1C), and, in doubtful cases, oral glucose tolerance tests. Patients that met American Diabetes Association Dm diagnostic criteria for 2011 (HbA1C ≥ 6.5%, fasting plasma glucose ≥ 126 mg/dL, or 2 h plasma glucose ≥ 200 mg/dL in the oral glucose tolerance test) [35] were excluded from this study. Cognitive functions were assessed by a neuropsychologist experienced in neurodegenerative disorders using the Mini-Mental Status Exam (MMSE) [36] or Montreal Cognitive Assessment (MoCA) test [37]. Patients with dementia were also excluded from this protocol.

From the 41 patients initially screened for this study, 2 were diagnosed with T2DM at the screening stage, 1 withdrew consent for participation, and 3 were then excluded due to insufficient data from CGM (only data from patients whose CGM time was longer than 75% were analyzed). The control group consisted of individuals who met the same inclusion and exclusion criteria except for the diagnosis of PD. The study protocol did not differ for these patients except for PD-specific testing. Control patients were recruited mainly from the participants’ families, e.g., spouses, as they had a similar age, lifestyle, and diet.

### 2.2. Enrollment Visit

Participants reported to this study on the day of enrollment in a fasting state and were also instructed to report in the OFF state (no dopaminergic medications on the day of the visit or in the evening of the previous day). A neurological examination in the OFF state was performed and patients were assessed using the Movement Disorder Society Unified Parkinson’s Disease Rating Scale part III (MDS-UPDRS p. III) [38]. Laboratory tests were performed for some patients; however, most patients had a full range of tests performed at the time of screening to limit unnecessary blood draws. The participants then took their standard morning dose of dopaminergic medication and were reassessed one hour after taking the medication (ON state), including an assessment using MDS-UPDRS p. III. Biometric data were collected, including height and body mass, and the body mass index (BMI) was calculated [39]. Patients were also familiarized with the Freestyle Libre system, the principles of taking measurements, and the scales they should use during monitoring.

### 2.3. Continuous Glucose Monitoring (GCM) Phase

During this phase, patients returned to their homes and were required to actively scan the Freestyle Libra sensor every 6 h or less using a specially provided reader or an application installed on their mobile device to ensure complete data collection. Participants were also given the researchers’ phone numbers to contact in case of problems with the system. If the sensor was accidentally removed and the participant wished to continue in this study, an additional sensor was provided. Participants were instructed not to change their usual behavior or diet. Participants were also asked to keep a food diary for at least two days and to monitor physical activity. The study protocol is summarized in Figure 1.

### 2.4. Study Assessment Measures

#### 2.4.1. Clinical Examinations (Neurological Assessment and Anthropometric Data)

The motor symptoms of PD were assessed using MDS-UPDRS p. III performed in the OFF and ON states; participants were also rated with the Hoehn–Yahr scale [40]. Anthropomorphic data, including height, body mass, and BMI, were collected. Patients were evaluated for the presence of autonomic dysfunction, and the severity of autonomic dysfunction was assessed using the Scale for Outcomes in Parkinson’s disease for Autonomic symptoms (SCOPA-AUT) [41].

The presence of a significant autonomic disorder was defined as follows:–Neurogenic orthostatic hypotension—manifested by a drop in arterial blood pressure of ≥20/10 mmHg after 3 min of standing or the occurrence of problems in the form of fainting, lightheadedness, or pain in the subscapular area and neck (coat hanger pain), and postprandial hypotension manifested by postural symptoms or fatigue 30–60 min after a meal.–Urinary tract involvement—urinary retention (frequent urination, urgency, urinary incontinence, voiding dysfunction).–Erectile dysfunction in men—symptoms must not be due to urological causes.–Gastrointestinal involvement—constipation, delayed bowel movement.–Sweat secretion disorders.

#### 2.4.2. Laboratory Parameters

The laboratory parameters included the fasting glucose levels, glycated hemoglobin (HbA1c), insulin, and lipid profiles. Assessments of renal and thyroid function and oral glucose tolerance tests (for selected patients) were carried out only for screening purposes, and these results were not analyzed, because only patients with normal results were included in this study.

#### 2.4.3. CGM Using the Freestyle Libre Sensor

The data from 14 days of CGM using the Freestyle Libre sensor include the average glucose, average glucose by time of day, glucose management index (GMI), approximation of HbA1c based on the average glucose from CGM, glucose variability, time in range (TIR, 70–180 mg/dL), time in tight range (TITR, 70–140 mg/dL), and hypoglycemic and hyperglycemic events. Only individuals with data for at least 75% of the monitoring period were included in the analysis. During the monitoring phase, participants were asked to keep food diaries (for at least 2 days) and to monitor physical activity, but due to the low compliance and variable quality of the data, these data were not suitable for further analysis.

### 2.5. Statistical Analysis

The statistical analysis was performed with the Statistica 13.3 software system (TIBCO Software Inc., Palo Alto, CA, USA, (2017). http://statistica.io). The quantitative variables are presented as the arithmetic mean and standard deviation (normally distributed variable) or median and interquartile range (variables with skewed distribution). The qualitative variables are presented as absolute values and percentages. The normality of distributions was assessed with the Shapiro–Wilk test.

Due to a lack of confirmation of a normal distribution in the analyzed groups, the intergroup differences for the quantitative variable were assessed with the Mann–Whitney U-test, Fisher’s exact test, or chi-square test. A *p*-value less than or equal to 0.05 was considered statistically significant. Odds ratios (ORs) with a 95% confidence interval (CI) and *p*-values were obtained using linear regression. The final predictive model for T2DM was fitted using the forward stepwise selection method. The significance level was set at *p* < 0.05.

### 2.6. Ethical Approval

This study received consent from the Bioethics Committee of the Medical University of Silesia under Resolution No. BNW/CBN/0052/KB1/32/III/22 of 12 July 2022 and the Resolution on Extension No. BNW/NWN/0052/KB1/32/IV/22/23 of 14 March 2023.

## 3. Results

### 3.1. Glycemic Characteristics of the PD Study Group Compared with the Control Group

A total of 55 individuals—35 diagnosed with PD and 20 healthy controls matched in terms of their age and gender distribution—were analyzed in this study. No difference between the groups was observed in terms of their average glucose, HbA1c, or daily glycemic variability (Table 3). However, a tendency toward lower glucose values in the morning was observed in the PD group. Statistically significantly lower average glucose levels were observed in this group from 0:00 a.m. to 9:00 a.m. Individuals in the control group demonstrated statistically significantly higher average glucose levels in the afternoon between 3:00 p.m. and 9:00 p.m. Individuals with PD also had reduced TITR compared to controls (92.5% vs. 94.3%, *p* = 0.0431). The groups did not differ in terms of age or gender. Detailed data are summarized in Table 3.

### 3.2. Characteristics of the Group of PD Patients

The PD group consisted of 35 individuals diagnosed with PD, comprising 15 females (42.9%) and 20 males (57.1%), aged 67.9 ± 9.4. The median Hoehn–Yahr score was 3 [IQR 2.5–3]; the median MDS-UPDRS p. III OFF was 51, interquartile range (45–60); and the median MDS-UPDRS p. III ON was 25, interquartile range (20–32). The group was assessed for the occurrence of autonomic disorders. Among the PD patients, 12 (32.3%) had urinary dysfunction, 9 (25.7%) had orthostatic hypotension, and the median score in the SCOPA-AUT was 17.5, interquartile range (8–28). For further analyses, the group was divided into patients with and without significant autonomic disorders. Detailed data are presented in Appendix A.

### 3.3. Autonomic Disorders and Clinical Features

The group of patients with significant autonomic disorders was older (66 vs. 72 years; *p* = 0.0361), but there were no differences in terms of gender, disease severity, BMI, motor symptoms, or treatment. Detailed data are summarized in Appendix A.

### 3.4. Two-Week Glycemia Monitoring in PD Individuals With and Without Significant Autonomic Disorders

The group of individuals with autonomic disorders was characterized by a much greater glycemic variability (14% vs. 20%; *p* < 0.001) over the 14-day CGM assessment period. Individuals with autonomic disorders were also characterized by a tendency toward lower average glucose values in the morning between 03:00 and 09:00. Individuals without autonomic disorders had a greater TIR, 70–180 mg/dL (98% vs. 94%; *p* = 0.0138), and a greater TITR (97% vs. 92%; *p* = 0.0345). Detailed data are summarized in Table 4.

### 3.5. Laboratory Results in Individuals With and Without Autonomic Disorders

In the comparative analysis of both groups, no differences were observed in fasting glucose, glycated hemoglobin, or endogenous insulin levels. Individuals without autonomic disorders had a more unfavorable lipid profile with higher LDL (112.8 mg/dL vs. 80.7 mg/dL; *p* = 0.0089) cholesterol levels and lower HDL cholesterol levels (60.7 mg/dL vs. 74.1 mg/dL; *p* = 0.0387). Detailed data are summarized in Appendix A.

### 3.6. Identification of Factors That May Predispose Patients to Increased Glycemic Variability

A linear regression model was built to identify factors predisposing patients to more significant glucose variability. HbA1C, SCOPA-AUT score, orthostatic hypotension, and micturition disorders were identified as predictors of more significant glycemic variability. Detailed results are presented in Table 5. The regression analysis indicates the value of the presence of autonomic disorders in predicting greater glycemic variability. This applies both to the global score of the SCOPA-AUT and, in particular, to the occurrence of orthostatic disorders and urination disorders, which was associated with higher glycemic variability. This effect has not been confirmed in other autonomic disorders. In laboratory tests, HbA1c has been shown to be useful in predicting greater glycemic variability.

## 4. Discussion

In this study, we found that autonomic disorders in PD increase the risk of glycemic dysregulation with greater risks for orthostatic hypotension and urination disorders. It should be emphasized that the occurrence of glycemic dysregulation in the group of individuals with autonomic disorders was not associated with traditional risk factors for the development of metabolic syndrome; for example, no differences were found in terms of BMI. The group without autonomic disorders had a more favorable lipid profile with higher HDL cholesterol and lower LDL cholesterol levels. Similar results were obtained in the previous retrospective study on the occurrence of prediabetic conditions in PD in relation to atypical Parkinsonism, where the higher incidence of glucose intolerance in PD was not associated with risk factors for the development of metabolic syndrome [30]. It should be emphasized, however, that the group of individuals with autonomic disorders was characterized by higher age, which is a known risk factor for prediabetes and T2DM, and a longer duration of the disease may result in a more frequent occurrence of autonomic disorders in the clinical picture of PD. There is some evidence suggesting a protective role for levodopa against the development of T2DM [42], but our study did not demonstrate the effect of treatment modalities on glycemic profiles. However, it should be emphasized that the study group was quite small and heterogeneous regarding the therapy used.

In the analysis of daily trends, a greater variability in CGM glucose levels and tendency toward lower morning glucose levels was observed in PD compared with controls, and this tendency was also more pronounced in individuals with autonomic dysfunction. More glycemic variability was also observed in people with autonomic disorders. This may be particularly relevant for individuals with PD and orthostatic hypotension, which is also most severe in the morning, and the co-occurrence of lower average glucose levels may exacerbate the clinical symptoms associated with morning hypotension. Other studies have indicated a high incidence of glucose intolerance among people with PD, reaching 50–80% [43]. However, a more recent study by Marques et al. from 2018 estimated the prevalence of glucose intolerance at 20%, which is more similar to the results from the general population [25]. Our study did not show any differences in glycemic variability between individuals from the control group and those with PD, which may indirectly indicate glycemic intolerance. Patients diagnosed with Parkinson’s disease were also characterized by shorter TIR and TITR compared to the control group; moreover, the coexistence of autonomic disorders in the group of patients with PD was also associated with shortened TIR and TITR, which may also indicate a greater tendency toward the dysregulation of glycemic metabolism in this group. It should also be noted that the differences in TIR and TITR were very similar, which does not confirm the usefulness of narrowing the glycemic range in monitoring these patients. No other studies to date have provided data that are directly comparable with these observations. However, our data may suggest the existence of a subpopulation of individuals with PD who are particularly vulnerable to the development of glycemic disorders. This fits with the ongoing debate about body-first versus brain-first PD [44]. In body-first PD, Lewy bodies can be found in the enteric nervous system eight years before the substantia nigra pars compacta [45]. Lewy pathology progresses via the nigro-vagal pathway [46]. Patients with body-first PD develop gastrointestinal symptoms and autonomic dysfunction at an earlier stage of the disease [44]. In the context of the present study’s outcomes, this may indicate that this subpopulation is more susceptible to greater glycemic variability. Researchers comparing the brain-first and body-first PD populations are encouraged to assess the incidence of glycemic disturbances in these groups.

It should also be noted that individuals with PD should be closely monitored for glycemic disorders, as the coexistence of diabetes and PD negatively affects the course of the disease [43,47,48]. T2DM is known to be associated with worse axial symptoms [49,50] and cognitive impairment [50,51]. Good glucose control is associated with a more favorable PD outcome [52]. According to the results of the current study, assessing individuals with PD for autonomic disorders may help select those who require screening for dysglycemia. Both the occurrence of individual disorders and a higher SCOPA-AUT score are predictive factors for increased glycemic variability, which may indicate an increased risk of developing T2DM. Among the laboratory tests that are diagnostic for T2DM, our data show the value of determining HbA1c in patients with PD. Recent studies showed that elevated HbA1c at ≥42 mmol/mol is an independent predictor of the development of PD [53] and is associated with worse severity of motor and cognitive function [54]. A study by Marques et al. [25] examining glucose control in individuals with PD without T2DM used a 75 g oral glucose tolerance test to demonstrate impaired glucose control in PD individuals. Similarly, in our study, we confirmed that greater glucose variability, which may indicate greater glycemic intolerance in individuals with PD, was related to the severity of autonomic disorders.

Given the complexity of factors that can lead to PD and T2DM, it is difficult to determine which factors are the most important determinants of the development of PD and T2DM [6,19]. In this study, we focused on clinical factors that may help to identify individuals with PD at risk of T2DM. Further studies on the common mechanistic pathways between PD and T2DM are needed to identify new therapeutic targets for both chronic diseases. Some of the available research suggests that the use of metformin [42,55,56], thiazolidinediones and GLP-1 receptor agonists [57,58,59,60], or DPP4 inhibitors [60,61,62] may reduce the risk of developing PD in people with T2DM. The exact mechanism linking insulin regulation and glucose metabolism with neurodegeneration remains unclear [6,15,19].

This study has several limitations that should be mentioned. Firstly, it was conducted in a relatively small group from the Polish population, and most of the individuals with PD were at a moderately advanced stage of the disease (2–3 on the Hoehn–Yahr scale), which may make it impossible to generalize the conclusions to the general population. Further multicenter research on PD patients at different stages of the disease, in different subtypes of PD, and exploring the influence of dopaminergic medications on glycemic profile are needed to validate these findings. Objective biomarkers such as functional neuroimaging or measures of alpha synuclein should be used. Secondly, in this study, we could not measure daily physical activity and dietary behavior as the study protocol relied on patients’ self-reported data. Further study involving objective measures (e.g., actigraphy for physical activity) should be performed to assess the influence of these factors on glycemic profile.

## 5. Conclusions

In this study, we observed a trend toward low morning blood glucose levels among people with PD. Autonomic disorders predispose such patients to the occurrence of more profound glycemic disorders with greater glucose level variability and a higher risk of morning glycemic drops. This may be of particular clinical importance for patients with orthostatic hypotension and may potentially worsen the disorder. Particular diagnostic vigilance should be maintained for glycemic disorders among patients with autonomic disorders.

## Figures and Tables

**Figure 1 healthcare-12-02462-f001:**
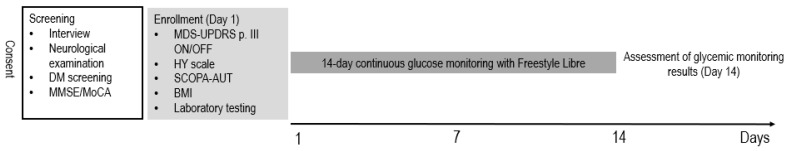
Outline of study protocol. DM—diabetes mellites, MDS-UPDRS—Movement Disorder Society Unified Parkinson’s Disease Rating Scale, HY—Hoehn–Yahr, SCOPA-AUT—Scale for Outcomes in Parkinson’s disease for Autonomic symptoms, MMSE—Mini-Mental State Examination MoCA—Montreal Cognitive Assessment, BMI—body mass index.

**Table 1 healthcare-12-02462-t001:** Potential pathophysiological links between PD and T2DM. PD—Parkinson’s disease, DM—diabetes mellites [8,14,15].

Potential Pathophysiological Links Between PD and T2DM	
Insulin signaling and dysregulation	Increase in neuroinflammation, promotion of apoptosis, decrease in insulin’s protective effect on brain
Insulin resistance and hyperglycemia	Decrease in dopamine transport, increase in neuroinflammation and oxidative stress, increase in methylglyoxal secretion that promotes α-synuclein aggregation
Oxidative stress and neuroinflammation	Increased loss of dopaminergic neurons, creation of α-synuclein-specific T cells, microglia activation
Protein aggregation	Cross-pollination between islet amyloid polypeptide (IAPP) and α-synuclein, amylin secretion increases α-synuclein formation and tau phosphorylation
Autonomic disorder	Promotion of impaired insulin secretion and regulation

**Table 2 healthcare-12-02462-t002:** Inclusion and exclusion criteria of this study. DBS—deep brain stimulation.

Inclusion Criteria	Exclusion Criteria
Diagnosis of Parkinson’s disease	Diagnosed diabetes (known from the interview or diagnosed during recruitment to this study)
Age over 18	Taking medications that may affect glycemic levels
Consent to participate in this study	Condition after DBS implantation
	Modification of Parkinson’s disease therapy within one month before the examination
	Patient with acute illness
	Chronic kidney disease stage above G4
	Diagnosed thyroid disease or other endocrine disease associated with glycemic disorder

**Table 3 healthcare-12-02462-t003:** Comparison of individuals with PD and control group. Statistical analyses were performed using Student’s *t*-test, the Mann–Whitney U-test, Fisher’s exact test, or the chi-square test. GMI—glucose management indicator; HbA1c—hemoglobin A1C; TIR—time in range 70–180 mg/dL; TITR—time in tight range 70–140 mg/dL; * approximation of HbA1c level based on the average glucose from continuous glucose monitoring; ** percentage of monitoring period with blood glucose data available.

	Control Group N = 20	PD N = 35	*p*
Age, years	66.3 ± 8.5	67.9 ± 9.4	0.2689
Gender			0.7787
Male, %	8 (40)	15 (42.9)	
Female, %	12 (60)	20 (57.1)	
Glucose management indicator (GMI) *	5.7 [5.55–5.7]	5.5 [5.4–5.8]	0.8473
Glucose variability	16 [13–17]	16 [13–18]	0.1171
Time in glucose ranges			
(% of the day)			
>250 mg/dL	0 [0–0]	0 [0–0]	0.4153
181–250 mg/dL	0 [0–0.6]	0.7 [0–3]	**0.0376**
70–180 mg/dL (TIR)	99 [98,5–100]	97 [94–99]	**0.0319**
54–69 mg/dL	0 [0–1]	0 [0–1]	0.5711
<54 mg/dL	0 [0–0]	0 [0–0]	0.8936
70–140 mg/dL (TITR)	94.3 [92–97]	92.5 [90–96]	**0.0431**
Average glucose concentrations by hours of the day (mg/dL)			
0:00–2:59	96.5 [87.5–102]	87 [81–96]	**0.0104**
3:00–5:59	96.5 [89.5–103]	87 [82–97]	**0.0422**
6:00–8:59	102.5 [97–109.5]	94 [88–103]	**0.0082**
9:00–11:59	106 [100.5–111]	102 [93–109]	0.1221
12:00–14:59	106 [103–112]	103 [100–110]	0.2032
15:00–17:59	110.5 [104–124]	102.5 [92–108]	**0.0126**
18:00–20:59	111.5 [101.5–119.5]	102 [92–108]	**0.0098**
21:00–23:59	105 [95–109]	99 [91–106]	0.3585
Time sensor active ** (%)	91 [84.5–98]	80.5 [75–87]	**0.0021**
Average glucose (mg/dL)	102 [96–108]	96 [90–105]	0.0943
HbA1c (%)	5.6 [5.5–5.8]	5.5 [5.3–5.7]	0.5432

**Table 4 healthcare-12-02462-t004:** The 14-day CGM metrics in PD individuals with and without significant autonomic disorders. Statistical analyses were performed using the Mann–Whitney U-test, Fisher’s exact, or the chi-square tests. TIR—time in range 70–180 mg/dL; TITR—time in tight range 70–140 mg/dL; * approximation of HbA1c level, based on the average glucose from continuous glucose monitoring; ** percentage of monitoring period with blood glucose data available.

	Without Autonomic Disorders	With Autonomic Disorders	*p*
Glucose management indicator (GMI) *	5.6 [5.4–5.8]	5.5 [5.4–5.8]	0.8911
Glucose variability	14 [12–16]	20 [18–21.5]	**<0.0001**
Time in glucose ranges			
(% of the day)			
>250 mg/dL	0 [0–0]	0 [0–0]	0.9727
181–250 mg/dL	0.2 [0–2]	1 [0–6]	0.1901
70–180 mg/dL (TIR)	98 [96–100]	94 [90–98]	**0.0138**
54–69 mg/dL	0 [0–0]	0 [0–5]	0.3361
<54 mg/dL	0 [0–0]	0 [0–0]	0.3361
70–140 mg/dL (TITR)	97 [95–98]	92 [89–95]	**0.0345**
Average glucose concentration	97 [91–105]	92 [89–104]	0.5149
Average glucose concentrations by hours of the day (mg/dL)			
0:00–2:59	87 [85–99]	85 [72–95]	0.1114
3:00–5:59	91 [85–99]	82 [73–96]	**0.0361**
6:00–8:59	96 [89–105]	92 [76–101]	**0.0467**
9:00–11:59	102 [96–106]	102 [90–114]	0.9823
12:00–14:59	102 [100–110]	105 [99–110]	0.6812
15:00–17:59	103 [100–106]	100 [96–115]	0.9727
18:00–20:59	103 [95–108]	100 [90–112]	0.4715
21:00–23:59	99 [91–106]	99 [90–106]	0.6562
Time sensor active ** (%)	80 [75–83]	83 [75–88]	0.2864

**Table 5 healthcare-12-02462-t005:** Factors that may predispose to increased glycemic variability. Linear regression results on predictive factors for higher glucose variability.

	OR	95% CI		*p*
SCOPA-AUT	0.23326	−0.0869	0.24761	**0.041818**
HbA1c	3.75942	3.0571	10.67829	**0.001234**
Orthostatic hypotension	−2.10791	−4.3988	−0.02302	**0.047854**
Urinary dysfunction	−1.19778	−2.3793	−0.64355	**0.045005**

OR—odds ratio; CI—confidence interval; SCOPA-AUT—Scale for Outcomes in Parkinson’s disease for Autonomic symptoms.

## Data Availability

The data presented in this study are available on request from the corresponding author. They are not publicly available due to privacy restrictions.

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
