# Peer review of "Glucose Metabolism Disorders and Parkinson’s Disease: Coincidence or Indicator of Dysautonomia?"

_healthcare, 2024, doi:10.3390/healthcare12232462_

Round 1

Reviewer 1 Report

Comments and Suggestions for Authors

This study aimed to analyze the incidence of glycemic dysregulation in individuals with Parkinson’s Disease (PD) and to identify clinical factors that may predispose them to metabolic disturbances. Although the research is relevant, original, and makes a contribution to this area, there are several limitations and biases that temper my enthusiasm. I offer the following suggestions and questions to improve the application of these findings:

Main Questions: 1st) Given that PD is age-related and most patients are sedentary, a high incidence of diabetes in this population is expected. In my view, diabetes is more closely related to aging and sedentarism than to PD itself. 2nd) The PD group consists of 35 participants, while the control group has only 20. This discrepancy is illogical, as recruiting control individuals is typically easier than recruiting PD patients. To confirm that diabetes is more closely related to PD than to age and sedentarism, both groups should be matched for age, sex, and lifestyle factors, which does not appear to have been a priority for the authors. 3rd) Avoid  self-citation; there are three references by Chmiela & Gorzkowska cited in this study.

Title: While the title captures the main idea of the study, it is lengthy and may not grab readers’ attention effectively. Given the vast number of studies published daily, I suggest the authors make it more concise and appealing.

Abstract: Like the title, the abstract requires revision. Some points indicate the need for greater care. For instance, the term “Parkinson’s Disease” should appear in full before the abbreviation “PD” is first used. Additionally, what is SCOPA-AUT? The methods section in the abstract is weak and could benefit from clarification. Language quality also requires improvement.

Keywords: Although not mandatory, I recommend using MeSH terms for keywords.

Introduction: Avoid citing too many references in short sentences (e.g., rows 39-41). Authors should clarify why they included 2 references from their group (references 9 and 27). Both studies appear to be retrospective; since many studies have been published on this topic, I suggest excluding these to avoid self-citation. Consider including recent non-retrospective studies that might contribute more.

Materials and Methods: This section is the weakest part of the study. Information appears scattered, with limited detail. Instruments are listed without explanation, and selection criteria are only presented in a table, without a rationale for their choice. There is no sample size calculation, no explanation for selecting 20 controls, and no details about on/off medication evaluations. In my view, this crucial part of the study was not properly explored.

Results: What do “Title 1” and “Title 2” mean in Table 2? Does Table 3 refer only to PD patients, or does it include controls? LEDD information should be detailed in the methods section. I recommend reducing the number of tables; excessive tables lacking a specific focus may lead readers to disengage from the study.

Discussion: The discussion section is generally fine. However, merely stating that “the study protocol could not objectively assess factors influencing glucose levels, such as physical activity and dietary behavior” is insufficient. Why not ask patients about physical activity, dietary habits, and lifestyle? Simply stating this as a limitation is inadequate.

Conclusion: Acceptable as written.

References: Avoid self-citation. I am uncertain if two retrospective studies are essential for the reference list.

Data Availability Statement: I recommend including the data in a supplementary file.

Comments on the Quality of English Language

The English could be improved to more clearly express the research.

Author Response

Dear Reviewer, 1.

Thank you for your detailed and insightful review of our work. I would like to thank you for all your comments and suggestions that helped us improve the quality of our work.

This study aimed to analyze the incidence of glycemic dysregulation in individuals with Parkinson’s Disease (PD) and to identify clinical factors that may predispose them to metabolic disturbances. Although the research is relevant, original, and makes a contribution to this area, there are several limitations and biases that temper my enthusiasm. I offer the following suggestions and questions to improve the application of these findings:

Main Questions:

Comment 1:1st) Given that PD is age-related and most patients are sedentary, a high incidence of diabetes in this population is expected. In my view, diabetes is more closely related to aging and sedentarism than to PD itself.

Response 1: The discussion about the interdependence of DM and PD is currently very lively, as evidenced by the number of publications published in recent years on this topic. It may be that the high prevalence of prediabetes and diabetes is due to a common risk factor (mainly age) and a decline in motor functions, favors the development of risk factors for metabolic syndrome. However, epidemiologic data suggest that this comorbidity cannot be clearly explained by the above-mentioned aspects. In our previous study cited in this paper (ref. 27), we evaluated the prevalence of prediabetes in PD and a large group of patients with atypical parkinsonism, interestingly PD (characterized by less significant disability), better results regarding BMI, lipid profile, etc., was characterized by a worse profile of glycemic metabolism. Patients with MSA were characterized by a similar (probably worse, but due to the size of the group it was not possible to confirm this statistically) glycemic profile. The frequency of abnormal fasting glycemia was similar to the frequency of autonomic dysfunction. There is no similar study performed (especially progressive as, as atypical parkinsonisms are rare disorders characterized by very rapidly developing disability). These results were one of justification for present study. Furthermore, other studies done in this field also supports this interdependence what led to clinical trials of antidiabetic medications in PD, nevertheless more data is needed to fully understand this relationship. Resent clinical trials on GLP1 agonists showing promising result, nut it is still unclear if improvement is not only related with optimalization of dopamine metabolism or GLP1 agonist have some disease modifying potential.

The method used in this study provides incomparably more data on glycemic metabolism - instead of individual measurements, each patient had a profile created from tens of thousands of glycemic measurements taken through 2 weeks period. This is the first application of this method in a group of patients with Parkinson's disease and provides new information on the glycemic profile in this group of patients. In introduction, we also add a brief summary in form of table that show potential pathophysiological link between DM and PD.

Comment 2: 2nd) The PD group consists of 35 participants, while the control group has only 20. This discrepancy is illogical, as recruiting control individuals is typically easier than recruiting PD patients. To confirm that diabetes is more closely related to PD than to age and sedentarism, both groups should be matched for age, sex, and lifestyle factors, which does not appear to have been a priority for the authors.

Response 2: In designing this study, we wanted to see not only whether PD patients differ from controls in terms of their glycemic profile, but also whether autonomic dysfunction has an impact on the occurrence of these disorders, so we wanted to have comparable groups of patients with and without autonomic dysfunction. Since we received funding for a limited number of sensors (a patient could require more than one sensor due to accidental removal, etc.).

The control group was age and gender matched, and they were also recruited from the families of the PD patients (because they have a similar lifestyle and dietary behavior). Patients were also instructed not to change their habits during the observation period. The study protocol also asked patients to complete food diaries and report physical activity, but the quality of the data we received was not suitable for analysis (no information or for example information like “I ate breakfast”, and they were instructed how to do it properly), only few patients were dedicated to completing this part. Since we encounter this problem, we add a comment in limitation about necessity of base these measures on objective data (e.g. actigraphy for physical activity, in fact in original protocol we intended to use actigraphy, but we did not receive funds for this part). I think this is an important consideration for those trying to replicate this study in the future and learned from our experience. We rework methodology, limitation to better describe these aspects.

Comment 3: 3rd) Avoid  self-citation; there are three references by Chmiela & Gorzkowska cited in this study.

Response 3: There are two citations of our previous works:

Ref 27. – as partially explained before this work is important because it involved large cohort of atypical parkinsonism and compared prevalence of impaired fasting glycaemia between PD and PSP. This results indirectly suggest role of autonomic dysfunction in synucleinopathies as potential factor influencing glucose profile in those patients (and it is unrelated to classical metabolic risk factors). I am not aware of similar study done on similar or larger cohort. Despite retrospective nature this work provides novel data. The data presented in this paper were also distinguished (despite the retrospective nature) at the IAPRD meeting in 2022.

Ref. 9 I can support the statement supported by the second citation with other sources, but these are not studies conducted in Poland or Central Europe, however, to avoid accusations of unethical citation, I will remove this reference (even though these results characterize the studied population).

Comment 4: Title: While the title captures the main idea of the study, it is lengthy and may not grab readers’ attention effectively. Given the vast number of studies published daily, I suggest the authors make it more concise and appealing.

Response 4: Title was shortened to: “Glucose metabolism disorders and Parkinson's disease - coincidence or indicator of dysautonomia?”

Comment 5: Abstract: Like the title, the abstract requires revision. Some points indicate the need for greater care. For instance, the term “Parkinson’s Disease” should appear in full before the abbreviation “PD” is first used.

Response 5: Abstract was improved for better clarity, mentioned mistake was corrected.

Comment 6: Additionally, what is SCOPA-AUT? The methods section in the abstract is weak and could benefit from clarification. Language quality also requires improvement.

Response 6: Abstract were modified to better reflect study methodology, purpose of using SCOPA-AUT scale was explained.

Comment 7: Keywords: Although not mandatory, I recommend using MeSH terms for keywords.

Response 7: Keywords were checked, and now all are MeSH terms.

Comment 8: Introduction: Avoid citing too many references in short sentences (e.g., rows 39-41).

Response 8: Wherever it was suitable, they were split to indicate exact information they are referring to.

Comment 9: Authors should clarify why they included 2 references from their group (references 9 and 27). Both studies appear to be retrospective; since many studies have been published on this topic, I suggest excluding these to avoid self-citation. Consider including recent non-retrospective studies that might contribute more.

Response 9: explained in answer 3.

Comment 10: Materials and Methods: This section is the weakest part of the study. Information appears scattered, with limited detail. Instruments are listed without explanation, and selection criteria are only presented in a table, without a rationale for their choice. There is no sample size calculation, no explanation for selecting 20 controls, and no details about on/off medication evaluations. In my view, this crucial part of the study was not properly explored.

Response 10: The methodology section has been almost completely rewritten to better describe the study protocol, its objectives, and the measures used in this work.

Comment 11: Results: What do “Title 1” and “Title 2” mean in Table 2?

Response 10: It is a mistake, it was corrected.

Comment 12: Does Table 3 refer only to PD patients, or does it include controls?

Response 11: It refers only to PD patients. Order of presentation was changed to be easier to follow now: first present comparison of PD and controls, then analysis od PD group. Table description was changed to clearly indicate PD group.

Comment 13: LEDD information should be detailed in the methods section.

Response 12: Methodology section was almost entirely redone, now describe study measures with much more details, including information on LEED.

Comment 14: I recommend reducing the number of tables; excessive tables lacking a specific focus may lead readers to disengage from the study.

Response 14: Table 2,3 and 6 were moved to supplementary files to improve manuscript readability.

Comment 15: Discussion: The discussion section is generally fine. However, merely stating that “the study protocol could not objectively assess factors influencing glucose levels, such as physical activity and dietary behavior” is insufficient. Why not ask patients about physical activity, dietary habits, and lifestyle? Simply stating this as a limitation is inadequate.

                Response 15: It is difficult to obtain reliable data from PD patients based on their own reports. In this study, the patients were not supervised during the observation period (they returned home with the sensor), and while from the researcher's point of view we were able to check that they were monitoring their glucose, there was no way to assess their diet or physical activity. Patients were asked to fill in food diaries and report physical activity, but most of them did not do so or did so in a way that did not allow analysis of this part of the data. In future studies, as mentioned before it is advisable to use objective tools (e.g. actigraphy). We have modified the methods and limitations to explain this complex problem that did not allow the evaluation of these aspects. The control group was recruited from the spouses of our patients, who had a similar lifestyle and diet (we also tried to select people with similar mobility).

Comment 16: Conclusion: Acceptable as written.

Response 16: Thank you!

Comment 17 References: Avoid self-citation. I am uncertain if two retrospective studies are essential for the reference list.

Response 17: explained in answer 3.

Comment 18: Data Availability Statement: I recommend including the data in a supplementary file.

Response 18: All data that could be publicly shared according to ethical standards and policy of Medical University of Silesia are disclosed in this manuscript.

Comment 19: The English could be improved to more clearly express the research.

Response 19: Manuscript was edited by MDPI provided English editing service.

Thank you again for taking the time to analyze our work, thank you for all your suggestions.

Sincerely yours,

Tomasz Chmiela on behalf of all Authors

Reviewer 2 Report

Comments and Suggestions for Authors

The manuscript by Chmiela and colleagues (ID: 3310964) aims to assess the incidence and progression of glycemic dysregulation in people with PD, also examining the role of autonomic dysfunction and clinical factors associated with metabolic disturbances in the absence of diagnosed diabetes. The authors conclude that there is a trend towards low morning glucose levels in people with PD, especially in the presence of autonomic disorders, which increase glycemic variability and the risk of morning hypoglycemia. They emphasize the need for careful monitoring of glycemic disturbances in patients with autonomic dysfunctions, as these may worsen conditions like orthostatic hypotension.

Generals comment

Considering the limited number of patients and controls included, it would be appropriate to highlight that this article describes a pilot study, a point that should be clearly emphasized in the text to better contextualize the results. Although it is mentioned in the discussion, it is recommended to also include this clarification in the study aims within the introduction. Additionally, an accurate review of the bibliography is recommended, prioritizing, where possible, the use of more recent references to ensure that the work reflects the latest findings in the medical field.

A complete review of the text is advised to correct any typographical errors. For example, in line 43, there is an error where “PD” is joined to the square bracket, which should be corrected to improve the readability and precision of the manuscript.

To make the introduction smoother and more accessible, it would be helpful to include a table or schematic summary that briefly lists the main factors common to PD and DM, along with potential risk factors. This approach could facilitate reader understanding by offering a clear and immediate overview of the points of connection between the two conditions.

Finally, it is suggested to revise the order of result presentation. It may be more useful to first provide an overview of the controls and patients (Table 4), followed by the tables describing the specific characteristics of PD patients. This sequence would help the reader grasp the general context before delving into specific details, improving the accessibility of the text.

It is also advisable to highlight the significant p's, the method is at the discretion of the authors; for example, you could put them in bold.

In conclusion, to improve the quality of the manuscript, a thorough re-reading and careful revision of the text is recommended. In particular, it is suggested to revise the "Materials and Methods" section to ensure that the protocols used are described with clarity and precision. Each step of the protocol should be enhanced with detailed information and relevant references to make each phase understandable, even to readers less familiar with the field. If well-known protocols are referenced, it is recommended to include the relevant data associated with these protocols. This will provide readers with all the necessary information to replicate the described experiment or to fully understand the reported data.

Specific comment

Line 40 “Epidemiological 39 data on the co-occurrence of PD and DM may, but are unclear”

It is advisable to specify what DM is, since it may not be immediately clear to less experienced readers that it refers to diabetes melliticus.

Line 73 “To our knowledge, this is the first study to include 14-day CGM monitoring in a population of people with PD.”

When acronyms such as CGM are used, it is right that the first time they are reported in the text they are written in full, even if in theory it is present in the abstract. It is more correct to report it in the text.

Line 258 “A study by Marques et al. [22] examining… without T2D, used a … in PD individuals.”

You must correct T2D with T2DM. It is also noted that DM or T2DM is sometimes used in the text, it is recommended to always use the same acronym to identify the pathology, unless reference is made to diabetes mellitus in general.

Line 91

The different Clinical Assessments that have been carried out should be a minimum explained in the materials and methods; moreover, they are devoid of any bibliographical reference.

Line 95

In the Materials and Methods it is reported that several tests have been carried out, but the results do not report the data of the MMSE and Moca. It is advisable to report the data.  

Author Response

Dear Reviewer, 2,

I would like to thank you for this insightful review of our work, and all comments and suggestions, that help us to improve quality of our work.

The manuscript by Chmiela and colleagues (ID: 3310964) aims to assess the incidence and progression of glycemic dysregulation in people with PD, also examining the role of autonomic dysfunction and clinical factors associated with metabolic disturbances in the absence of diagnosed diabetes. The authors conclude that there is a trend towards low morning glucose levels in people with PD, especially in the presence of autonomic disorders, which increase glycemic variability and the risk of morning hypoglycemia. They emphasize the need for careful monitoring of glycemic disturbances in patients with autonomic dysfunctions, as these may worsen conditions like orthostatic hypotension.

Generals comment

Comments 1. Considering the limited number of patients and controls included, it would be appropriate to highlight that this article describes a pilot study, a point that should be clearly emphasized in the text to better contextualize the results. Although it is mentioned in the discussion, it is recommended to also include this clarification in the study aims within the introduction.

Response 1: Thank you for this suggestion, we added clear explanation of studies’ limitation in studies aim at the end of introduction. We mentioned that study is first of its kind, and that this result need further validation in study performed in other canters. Nevertheless, method used in this study provides much more insight on glucose metabolism: glucose profile bade on thousands of measurements. We believe that this data adds new knowledge to the field, and we strongly encourage other researcher to use GCM in their research in movement disorders.

Comments 2: Additionally, an accurate review of the bibliography is recommended, prioritizing, where possible, the use of more recent references to ensure that the work reflects the latest findings in the medical field.

Response 2: Article was reviewed for more recent literature references.

Comments 3. A complete review of the text is advised to correct any typographical errors. For example, in line 43, there is an error where “PD” is joined to the square bracket, which should be corrected to improve the readability and precision of the manuscript.

Response 3: Thank you for pointing this out, we have carefully reviewed his text for typographical errors and corrected them.

Comments 4. To make the introduction smoother and more accessible, it would be helpful to include a table or schematic summary that briefly lists the main factors common to PD and DM, along with potential risk factors. This approach could facilitate reader understanding by offering a clear and immediate overview of the points of connection between the two conditions.

Response 4: Table was added to introduction that briefly summarized the potential pathophysiological links between PD and DM. This is just an overview as knowledge on this subject is very broad. There are few recently published reviews that try to summarize this knowledge.

Comments 5: Finally, it is suggested to revise the order of result presentation. It may be more useful to first provide an overview of the controls and patients (Table 4), followed by the tables describing the specific characteristics of PD patients. This sequence would help the reader grasp the general context before delving into specific details, improving the accessibility of the text.

Response 5: Order of data presentations was changed, thank you for this suggestion, I agree that is much more logical.  

Comments 6: It is also advisable to highlight the significant p's, the method is at the discretion of the authors; for example, you could put them in bold.

Response 6: Thanks for this suggestion, significant p-values are now in bold.

Comments 7: In conclusion, to improve the quality of the manuscript, a thorough re-reading and careful revision of the text is recommended. In particular, it is suggested to revise the "Materials and Methods" section to ensure that the protocols used are described with clarity and precision. Each step of the protocol should be enhanced with detailed information and relevant references to make each phase understandable, even to readers less familiar with the field. If well-known protocols are referenced, it is recommended to include the relevant data associated with these protocols. This will provide readers with all the necessary information to replicate the described experiment or to fully understand the reported data.

Response 7: The methodology section was almost completely redone, now it provides details about study protocol. I have also updated a figure outlining study protocol to better reflect each study phase.

Specific comment

Comments 8: Line 40 “Epidemiological 39 data on the co-occurrence of PD and DM may, but are unclear” It is advisable to specify what DM is, since it may not be immediately clear to less experienced readers that it refers to diabetes melliticus.

Response 8: We added information about DM, furthermore we provided some information about T2 development to introduce reader with term of prediabetes.

Comments 9: Line 73 “To our knowledge, this is the first study to include 14-day CGM monitoring in a population of people with PD.” When acronyms such as CGM are used, it is right that the first time they are reported in the text they are written in full, even if in theory it is present in the abstract. It is more correct to report it in the text.

Response 9: Thank you for pointing this, it was corrected.

Comments 10: Line 258 “A study by Marques et al. [22] examining… without T2D, used a … in PD individuals.” You must correct T2D with T2DM. It is also noted that DM or T2DM is sometimes used in the text, it is recommended to always use the same acronym to identify the pathology, unless reference is made to diabetes mellitus in general.

Response 10: Text was revied and abbreviation for type 2 diabetes mellites was unified to T2DM. DM is only used when it refers diabetes as a broader term.

Comments 11: Line 91 The different Clinical Assessments that have been carried out should be a minimum explained in the materials and methods; moreover, they are devoid of any bibliographical reference.

Response 11: Methodology section is now almost entirely rewritten, now provides much more details on scales, that were used (with proper citations).

Comments 12: Line 95 In the Materials and Methods it is reported that several tests have been carried out, but the results do not report the data of the MMSE and Moca. It is advisable to report the data.  

Response 12: Data on cognitive tests (MMSE or MoCA) were not included in the results section because we enrolled only patients without cognitive impairment who could comply with the study protocol. The results obtained are therefore the result of the patient selection method and are therefore not reported. We have made numerous changes to the Methods section to improve the clarity of the presentation of the study protocol.

Thank you again for all your effort in reviewing this manuscript.

Tomasz Chmiela on behalf of all Authors.

Reviewer 3 Report

Comments and Suggestions for Authors

In the article entitled “Glucose metabolism disorders and Parkinson's disease

- coincidence of morbidity or indicator of dysautonomia: a 14-day continuous glucose monitoring study” the authors identified the link between Parkinson's disease and glycemic dysfunction, where there is possibly a dysfunction of the nervous system affecting glucose regulation.

There are a few points that need to be addressed.

1.     The size of the sample, since it is a small, specific group from a population in Poland. This may not be a sample that is representative. It would be important to do a multicenter study.

2.     Most of the participants in the study are in stages 2-3 of the Hoehn-Yarhr scale, which is a restriction of interesting results that may occur in mild or more advanced stage of the disease. Studies have shown that glucose dysregulation is an early occurring event in Parkinson's disease.

3.     There is a lack of comparison of Parkinson's disease subtypes, such as body-first vs. brain-first.

4.     Variations by some type of physical activity and diet, which could influence glucose levels.

5.     There would be a lack of PET imaging studies, using 18F-fluorodeoxyglucose, which could tell us about glucose metabolism in the brain of Parkinson's patients.

6.     It would be interesting to compare different stages of Parkinson's disease with glucose and alpha-synuclein levels in blood samples.

The study is interesting, and the results obtained agree with other studies that have been carried out on glucose and Parkinson's disease. It can be considered a pilot study, which it is recommended to extend it in number of patients, multicenter, different stages of the disease and if possible, to try to measure alpha-synuclein.

Author Response

Dear Reviewer, 3

I would like to thank you the insightful and substantial comments. Thank you for all your suggestion, that help us to prepare this revised version of this manuscript.

In the article entitled “Glucose metabolism disorders and Parkinson's disease

- coincidence of morbidity or indicator of dysautonomia: a 14-day continuous glucose monitoring study” the authors identified the link between Parkinson's disease and glycemic dysfunction, where there is possibly a dysfunction of the nervous system affecting glucose regulation.

There are a few points that need to be addressed.

Comments 1. The size of the sample, since it is a small, specific group from a population in Poland. This may not be a sample that is representative. It would be important to do a multicenter study.

Response 1: I totally agree, this is the first study ever using continuous glucose monitoring in PD patients. This method gives us incomparably more insight into the patient's glucose profile than standard laboratory testing. This method is widely used in diabetology (especially in DM type 1). By publishing this paper, we also aim to raise interest in this method for studies in PD, and we hope that researchers from other centers will replicate our research. We have modified the manuscript to better describe the nature of this study and to encourage other researchers to explore this area. The methodology section was almost entirely redone.

Comments 2.     Most of the participants in the study are in stages 2-3 of the Hoehn-Yarhr scale, which is a restriction of interesting results that may occur in mild or more advanced stage of the disease. Studies have shown that glucose dysregulation is an early occurring event in Parkinson's disease.

Response 2” Yes, we decided to enroll mainly patients in HY 2-3 because our interest was in the association of autonomic dysfunction with glucose metabolism profile. This is a continuation of our previous work (ref. 27) where we retrospectively showed that prediabetes was much more common in PD and MSA compared to PSP (which almost exactly matches the prevalence of autonomic dysfunction in these diseases and was unrelated to classical metabolic risk factors). Since autonomic symptoms are not often present in early PD and we wanted to have comparison of PD patients with and without dysautonomia and we decided to focus on those stages. We also did not enroll very advanced PD patients (in terms of mobility, e.g. we had patients with advanced PD treated with LCIG or CSAI, but they were in very good overall condition, with good symptom control), as we were comparing them with controls recruited mainly from their spouses to match in terms of mobility and lifestyle. We are improving the methods section to better explain this issue. Further research is needed to extend this protocol to early and advanced stages.

Comments 3: There is a lack of comparison of Parkinson's disease subtypes, such as body-first vs. brain-first.

Response 3: Sample size was unfortunately too small to run such analysis, there would be also methodological problems with defining these those group. Considering this we decide to focus on autonomic disorders. This is a great idea for further research.

Comments 4.     Variations by some type of physical activity and diet, which could influence glucose levels.

Response 4: We tried to evaluate this; it was planned to be based this on patients' questionaries that should be completed by patients on certain dates. Unfortunately, patients (since they were unsupervised during study), have not completed them with sufficient quality. Unfortunately, the results from this part of the study are not suitable for analysis. Originally, we planned to assess physical activity using actigraphy (this would give an objective result, independent of the patients' willingness), but unfortunately we did not receive funding. Therefore, I mentioned in the limitations that it is necessary to use objective measurements regarding these issues. The methodology section and the limitations have been modified to better describe this problem and to serve as a guide for future researchers.

Comments 5.     There would be a lack of PET imaging studies, using 18F-fluorodeoxyglucose, which could tell us about glucose metabolism in the brain of Parkinson's patients.

Response 5: Objective measures, compared with data on glucose would be great. We add this suggestion to manuscript.

Comments 6.     It would be interesting to compare different stages of Parkinson's disease with glucose and alpha-synuclein levels in blood samples.

Response 6: It would be a great next step in research, we add your suggestion to manuscript text.

Comments 7: The study is interesting, and the results obtained agree with other studies that have been carried out on glucose and Parkinson's disease. It can be considered a pilot study, which it is recommended to extend it in number of patients, multicenter, different stages of the disease and if possible, to try to measure alpha-synuclein.

Response 7: We are very encouraged that we grabbed your interest. Yes, this study is novel, and CGM was not performed by anyone, and these results should be validated by other researchers. Thank again for your time reviewing our work.

Best Regards,

Tomasz Chmiela on behalf of all Authors

Round 2

Reviewer 3 Report

Comments and Suggestions for Authors

I thank the authors for their replies. With the changes done to the manuscript, it was substantially improved.